



# Probabilistic modelling of the dependence between rainfed crops and drought hazard

Andreia F. S. Ribeiro[1], Ana Russo[1], Célia M. Gouveia[1,2], Patrícia Páscoa[1], Carlos A. L. Pires[1]

[1]Instituto Dom Luiz (IDL), Faculdade de Ciências, Universidade de Lisboa, 1749-016 Lisboa, Portugal
[2]Instituto Português do Mar e da Atmosfera, Lisboa, Portugal

*Correspondence to*: Andreia F.S. Ribeiro (afsribeiro@fc.ul.pt)

**Abstract.** Extreme weather events, such as droughts, have been increasingly affecting the agricultural sector causing several socio-economic consequences. The growing economy requires improved assessments of drought-related impacts in agriculture, particularly under a climate that is getting drier and warmer.   This work proposes a probabilistic model which
intends to contribute to the agricultural drought risk management in rainfed cropping systems. Our methodology is based on a bivariate copula-approach using Elliptical and Archimedean copulas, which application is quite recent in agrometeorological studies. In this work we use copulas to model joint probability distributions describing the amount of dependence between drought conditions and crop anomalies. Afterwards, we use the established copula models to simulate pairs of yield anomalies and drought hazard, preserving their dependence structure, to further estimate the probability of crop-loss. In the first step, we
analyse the probability of crop-loss without distinguishing the class of drought, and in a second step we compare the probability of crop-loss under drought and non-drought conditions. The results indicate that, in general, Archimedean copulas provide the best statistical fits of the joint probability distributions, suggesting a dependence among extreme values of rainfed cereal yield anomalies and drought indicators. Moreover, the estimated conditional probabilities suggest that the likelihood of crop-loss under dry conditions is higher than under non-drought conditions. From an operational point of view, the results aim to
contribute to the decision-making process in agricultural practices.

## 1 Introduction

Agriculture is one of the activities most directly influenced by climate variability (IPCC, 2012; Lesk et al., 2016) and particularly by extreme weather events (IPCC 2012). The latter are a major source of risk in agricultural systems, often entailing substantial crop yield losses (Bokusheva et al., 2016; Kogan et al., 2015; Saadi et al., 2015). Despite the constrains associated
with the application of certain governmental policies in the agricultural systems, the success of mitigating the consequences of climate extremes is largely dependent on the development of appropriate risk management strategies (Paredes et al., 2014; Quiroga et al., 2011). For this purpose, probabilistic information of the agricultural risk associated to certain meteorological conditions is currently a major requirement (Iglesias and Quiroga, 2007; Jayanthi et al., 2014; Madadgar et al., 2017),



particularly under the scope of the projected climate changes (Ferrise et al., 2011; Hernández-Barrera and Rodríguez-Puebla, 2017).

The management of agricultural drought risk is gaining further attention from researchers and stakeholders, particularly in regions dominated high precipitation variability and recurrent dry and warm episodes, such the Mediterranean region and in

particular the Iberian Peninsula (IP) (Martin-Vide and Lopez-Bustins, 2006; Sousa et al., 2011; Vicente-Serrano et al., 2014). Recent works have found significant negative trends of drought indexes in the IP, particularly in southern regions (Páscoa et al., 2017a; Sousa et al., 2011), and the expected declining of crop yields due to future warming conditions is being pointed out (Ferrise et al., 2011; Hernández-Barrera and Rodríguez-Puebla, 2017).

The assessment of yield variability based on crop and meteorological information is crucial for a more stable farmer income

and management (Reidsma et al., 2010). The recently developed drought index SPEI (Standardized Precipitation Evapotranspiration Index (Vicente-Serrano et al., 2010)) is found to be particularly suitable for agricultural drought applications in Mediterranean regions (Zampieri et al., 2017) and shows significant correlations with crop yields in the IP (Páscoa et al., 2017b; Ribeiro et al., 2018a). On the other hand, crop models describing the biological processes are one of the existing tools used to assess crop productivity. However, such models are limited in their ability to quantify the impact of

climate variability on crop yields over larger scales (Estes et al., 2013). In addition, the use of satellite-based data is increasing for agricultural purposes (Kogan et al., 2015; Rojas et al., 2011), and considerable correlations between remote-sensing of vegetation and crop yield are found in the IP (Gouveia and Trigo, 2008; Ribeiro et al., 2018a; Vicente-Serrano et al., 2006). Some studies have considered the use of different remote sensing drought indicators to account for different crop sensitivities to drought, such as to moisture and thermal conditions over the vegetative cycle (Bokusheva et al., 2016; Kogan, 2001; Ribeiro

et al., 2018a; Zarei et al., 2013). Moreover, the establishment of models for estimating crop yield under drought influence, using the combination of different drought indicators and different time-scales of drought occurrence, have shown an added value in the performance of the crop yield simulations over the IP (Hernandez-Barrera et al., 2017; Ribeiro et al., 2018a; Vicente-Serrano et al., 2006).

The modelling of crop yield variability under drought influence using statistical methods have been previously used to estimate

drought-related crop-losses (Kogan et al., 2015; Ribeiro et al., 2018a; Zampieri et al., 2017). Some authors have estimated crop yield probability distribution functions to find crop-specific risk levels and have applied Monte Carlo methods to generate large sample sizes of yield distributions over Mediterranean areas (Iglesias and Quiroga, 2007; Resco et al., 2010). At the country-level in Europe, Naumann et al. (2015) have developed drought damage functions using a single power-law dependence between drought severity and the associated damage. At a regional level in the IP, regression techniques

(Hernandez-Barrera et al., 2017; Hernández-Barrera and Rodríguez-Puebla, 2017; Ribeiro et al., 2018a) and artificial neural network (ANN) models (Ribeiro et al., 2018a) have been used to model response of rainfed winter cereal yields to drought conditions.

More recently, copula-based models have been applied for agricultural purposes, to model the dependence structures between crop yields and environmental conditions using joint distributions (Bokusheva et al., 2016; Li et al., 2015; Madadgar et al.,



2017). The concept of copulas is quite popular in financial risk modelling, and recently is becoming a valuable tool to model the risks associated to climate hazards, such as droughts (Ganguli and Reddy, 2012; Mirabbasi et al., 2012; Serinaldi et al., 2009). Based on the Sklar's theorem (Sklar, 1959) a copula approach "joins" the probability of drought occurrence and the probability of crop-losses caused by the drought event. A detailed description about the use of copulas is provided by Nelsen

5    (2013).

A major advantage of copula methods is the generation of joint distributions independently of their marginal distribution functions (Maity, 2018; Nelsen, 2013). Copula functions show a great flexibility in modelling the dependence between individual variables (such as crop yield and drought indicators) with complex relationships without making heavy assumptions. In addition, copula functions are adequate for modelling rare events in multivariate distributions and to generate large samples,

allowing to find the probability that individual variables will not exceed a certain extreme (tailed) value (Madadgar et al., 2017). A recent study by Madadgar et al. (2017) have produced probability distributions of rainfed crop yields in Australia under drought impacts based on copula-based techniques, using the Standardized Precipitation Index (SPI) and the Standardized Soil-moisture Index (SSI). For crop insurance purposes at the farm-level in Kazakhstan, Bokusheva et al. (2015) modelled the joint distributions of wheat yields and two satellite-based drought indices (Vegetation Condition Index (VCI)

and Temperature Condition Index (TCI)).

In this study, a copula-based approach is adopted to model the joint probability density function of crop yield and the drought conditions for probabilistic yield assessment, based on the data and empirical analysis previously considered in Ribeiro et al. (2018). This method allows to estimate the dependence structures between the probability distributions of crop yield and drought indicators using copula functions. The novelty and interest of this approach relates to the fact that this methodology

will allow to estimate the likelihood of crop-loss and compare the expected losses under drought conditions and non-drought conditions in the IP. This key question poses as a current demand with most interest to stakeholders, such as farmers and insurance companies, to mitigate agricultural drought risk over the major agricultural areas in the IP.

## 2 Data and methods

### 2.1 Study area and data

The exposure analysis performed by Ribeiro et al. (2018) allowed the identification of two clusters of provinces in the IP dominated by rainfed agricultural practices (Fig. 1), located approximately in the regions of Castilla-Léon (cluster 1 - northern region) and Castilla-La Mancha (cluster 2 – southern region). Given the suitability of using these two clusters for an agricultural drought analysis at the regional level, here we have considered the same area selection criteria: provinces with more than 50% of the territory occupied by agricultural areas and more than 50% of rainfed crops according to the CLC (2012) (for more

details please see Ribeiro et al. (2018)). For the same reason, and for sequential purposes, the crop and drought hazard data used in Ribeiro et al. (2018) have been incorporated in the present study to analyse the distributions of probabilities. Spatial averages of annual yield anomalies (t/ha) of barley and wheat were computed over the two clusters during the period of 1986-





2012, based on production (tons, t) and area (ha) information obtained from the Portuguese National Statistics Institute (INE) and the Spanish Agriculture, Food and Environment Ministry.

Drought conditions were investigated using the hydro-meteorological drought indicator SPEI and three satellite-based indices obtained from NOAA-AVHRR since 1981, namely the Vegetation Condition Index (VCI) (Kogan 1990), the Temperature

Condition Index (TCI) (Kogan 1995) and the Vegetation Health Index (VHI) (Kogan 1995). The monthly drought index SPEI gridded values, with spatial resolution of 0.5°, were computed based on precipitation and temperature values from the Climate Research Unit TS3.21 database (Harris et al., 2014), using a variety of time scales (1 to 12 months). The temporal multiscalar character of SPEI constitutes a significant advantage, allowing the characterization of different response times to drought conditions. In addition, the SPEI has been particularly suitable to monitoring drought conditions in the IP (Páscoa et al., 2017a;

Russo et al., 2017). The SPEI is a measure of climatic water balance $D$ (Eq. 1) given by the monthly difference between precipitation ($P$) and reference evapotranspiration ($ET_0$) (Vicente-Serrano et al., 2010),

$$D = P - ET_0 .$$  (1)

Following the work of Beguería et al. (2014), the $ET_0$ was determined based on the Hargreaves equation (Hargreaves and Samani, 1985). A log-logistic distribution was used for the statistical fitting of $D$ (Vicente-Serrano et al. 2010) accumulating

previous months. Further spatial averages were computed for each cluster of provinces.

While SPEI computation uses climatic water balance anomalies incorporating the role played by the evaporative demand on the occurrence of dry events (Vicente-Serrano et al., 2010), the remote sensing indices characterize the moisture, through the VCI (Kogan, 1995), the temperature induced stress through the TCI (Kogan, 1990) and health of vegetation, through the VHI (Kogan, 1997). The weekly global maps of VCI, TCI, and VHI were retrieved at 4km spatial resolution from NOAA's ftp

server (ftp://ftp.star.nesdis.noaa.gov/pub/corp/scsb/wguo/data/VHP_4km/geo_TIFF/). Some major advantages of remote sensing of vegetation involve the easy collection of data and coverage of very large areas, and the referred indices have been successfully considered for agricultural purposes (Bokusheva et al., 2016; Dalezios et al., 2014; Kogan et al., 2015). The VCI (Eq. 2) and TCI (Eq. 3) are expressed in terms of the Normalized Difference Vegetation Index (NDVI) and the Brightness Temperature (BT), and VHI is an average of the two in order to consider their combined effect of vegetation health (Eq. 4).

$$VCI = 100 \times \frac{NDVI - NDVI_{min}}{NDVI_{max} - NDVI_{min}}$$  (2)

$$TCI = 100 \times \frac{BT_{max} - BT}{BT_{max} - BT_{min}}$$  (3)

$$VHI = \left( \frac{VCI + TCI}{2} \right)$$  (4)

The selection of the most relevant drought indicator for each cereal and cluster was performed based on the largest absolute value of the standardized regression coefficients from the models developed in Ribeiro et al. (2018), in order to constitute pairs

of cereal yield anomalies and drought indicators (Table 1). First, a pool of the drought indicators better related with wheat and barley yield were chosen based on stepwise regression (95% confidence level), selecting the most statistically significant time scales and months of SPEI, together with the weeks of VHI, VCI and TCI. Afterwards, from the pool of selected drought





indicators, the one with largest contribution to the yield's variance was selected (Table 1). Consequently, for each cereal time series, the joint probability with drought conditions was estimated using one drought indicator. A major conclusion in Ribeiro et al. (2018) was that there are stronger relationships between remote sensing indices and cereal yield in the northern sector (cluster 1), and between SPEI and cereal yield in the southern sector (cluster 2). This selection of drought indicators highlights

that the influence of the climate conditions on agricultural activities varies according to the location, type of crop and the moment of the vegetative cycle. While cereals in cluster 1 are more influenced by the effects of temperature (TCI) and vegetation health (VHI) during late growth stages (weeks 23 and 22 correspond approximately to end of May and beginning of June, respectively for wheat and barley), cereals in cluster 2 are dominated by drought conditions described by SPEI much earlier, in the beginning of the intermediate growth stages (February and April with 5 and 1 month of time-scale, respectively

for wheat and barley). Table 1 resumes the data and the two top panels of Fig. 2 show the Empirical Cumulative Distribution Functions (ECDF) of the individual variables considered for the copula application. In summary, the data sample dimension is $n = 27$, corresponding to the annual values of the yield anomalies and the drought indicators during the selected weeks in the case of the TCI and VHI, and the selected timescales and months of SPEI, over the period 1986-2012.

## 2.2 Copula-based method

### 2.2.1 The concept of copula

Copula functions are powerful tools used to estimate the joint distribution between variables (Bokusheva et al., 2016; Madadgar et al., 2017; Zhang et al., 2011). The concept of copula was firstly introduced by Sklar (1959) to decompose a joint cumulative distribution function $F_{XY}(x, y)$ into two parts (Eq. 5): the marginal distribution functions $F_X(x) = u$ and $F_Y(y) = v$, and the copula $C$ describing the dependence part of the joint distribution,

$$F_{XY}(x, y) = C(u, v), \tag{5}$$

where the margins $u$ and $v$ are uniform on the interval [0,1] (Nelsen, 2013). This study adopts a bivariate modelling approach such that, for each pair $(X, Y)$ of cereal and drought indicator over each cluster we considered bivariate copula functions to estimate the joint probability distributions. Trivariate copulas have been proposed in the analysis of hydrological extremes (Afshar et al., 2016; Bezak and Brilly, 2014; Saghafian and Mehdikhani, 2014), but the development of higher dimensional

copulas exhibits very complex structures and further studies and evaluations are required. In comparison to high-dimensional copulas, the two-dimensional copulas involve much less computational cost and allows for more easily interpretable and illustratable relationships between the interval margins. For this reason, in the present study we restricted the analysis to the bivariate case using two-dimensional copulas, simplifying the results interpretation.

There is a range of copula families described in the literature which are able to estimate the dependence between the univariate

variables (Nelsen, 2013). The most commonly used copula families focus on the Archimedean and Elliptical classes (Maity, 2018). There are three Archimedean copulas particular popular given their simple functional form and their different patterns of dependence captures – Clayton, Gumbel and Frank – while there are two most popular Elliptical copulas derived from



Elliptical distributions – Gaussian and t-copula. These five copula functions are well-documented and have been employed in recent agrometeorological studies with a number of annual observations similar to our study (Bokusheva et al., 2016; Madadgar et al., 2017; Zscheischler et al., 2017). Table 2 summarizes the mathematical expressions of the referred copula functions considered in the present study.

An important concept for studying extreme events is the tail dependence, whose importance is more critical than the overall dependence structure for risk analysis (Bokusheva, 2014). The joint tail behaviour describes the amount of dependence in the corners of upper-right and lower-left quadrants (i.e. joint extreme events), and its representation depends on the type of copula (Nelsen, 2013). The Clayton, Gumbel and t-copula are able to capture tail dependence, whereas

Frank and Gaussian describe a joint symmetric structure without tail dependence. However, while Clayton and Gumbel copulas
capture an asymmetrical tail dependence, the students t-copula describe a symmetric tail dependence. The Clayton/Gumbel copulas capture greater dependence in the lower/upper tail suggesting greater probabilities of joint lower/upper extremes (i.e., lower/higher values of yield anomalies given lower/higher values of drought indicators).

### 2.2.2 Fitting of the copula functions

The estimation of the copula parameters can be performed using different methods based on maximum likelihood, such as
Maximum Likelihood Estimate (MLE), Inference Functions for Margins (IFM) or Canonical Maximum Likelihood (CML) (Maity, 2018). With MLE, both individual margins and copula parameters are estimated together, whereas with IFM the marginal parameters are first estimated individually. In this study the statistical inference of the copula functions is performed with the CML method, which stands for a nonparametric estimation of the margins. In this way, the individual variables were first transformed to the unit scale (pseudo-observations) using the kernel density estimator of the CDF, without making
assumptions about the marginal distributions (Fig. 2). The drawback of the shorter sample size is surpassed by the nonparametric estimation of the margins, which avoids heavy assumptions about their distributions, even when the available sample is rather small (Corder and Foreman, 2011; Fahr, 2017). The fitting of the bivariate copula functions was then applied to the pseudo-observations and the dependence parameters were estimated by means of maximum likelihood (Fig. 2). Figure 2 summarizes the main steps of the copula-based approach adopted in the present study. For a detailed description on fitting
methods please see Maity (2018).

The Akaike's Information Criteria (AIC) is frequently employed as a model selection tool in copula modelling (Li et al., 2015; Mirabbasi et al., 2012). Therefore, the selection of the best copula function for each pair of cereal and drought indicator was made based on the evaluation of AIC values calculated as AIC = –2 × (sum of log-likelihood) + 2 × (number of parameters) (Fig. 2). The copula function minimizing the AIC value was selected for each case. For verification purposes, the leave-one-
out cross-validated log-likelihood was also computed during the estimation of the parameters. This step was performed to confirm the reliability of the selected copula models and we found that, in general, the same functions are selected with both the AIC and the cross-validated log-likelihood criteria. For this reason, and given the wide use of the AIC, only the results for model selection based on the AIC will be presented.



### 2.2.3 Probability of Non-Exceedance and Conditional Probability of Non-Exceedance

After the estimation of the copula parameters, the established models are used to simulate 1000 pairs of uniformly distributed data (Fig 2). In the present study, let $F_{X_{sim}}(x) = u_{sim}$ denote the simulated distribution of yield and $F_{Y_{sim}}(y) = v_{sim}$ the simulated distribution of drought indicator. The data generation using the joint relationship preserves the dependence structure between the margins. The simulated data in the range [0, 1] is transformed back to the original scale using the kernel estimations of the inverse CDF, originating $X_{sim}$ and $Y_{sim}$, respectively.

First the copula simulations were used to estimate the risk of crop-loss in terms of the probability of not exceeding a threshold value of yield, i.e., Probability of Non-Exceedance (PNE) (Fig. 2). In this study we considered the threshold of minus one standard deviation $(-X_{std})$ of each cereal yield anomalies time-series, as we are focused in real losses of yield and not just values below the mean (Eq. 6).

$$PNE_{-X_{std}} = u_{sim}(-X_{std}) = Pr(X_{sim} \leq -X_{std}) \tag{6}$$

The PNE gives information about how likely the occurrence of a yield value below a certain threshold is. In other words, it gives the expected chance in percentage that the negative yield anomaly will not exceed (i.e. is not higher than) minus one standard deviation (-1 std).

Afterwards we have partitioned the simulated data points of $X_{sim}$ into those corresponding to drought (e.g. SPEI <= -0.84 (Agnew, 2000) and/or VHI <= 40 (Kogan, 2001)) and non-drought conditions (e.g. SPEI > -0.84 and/or VHI > 40) (Fig. 2). The respective cumulative distribution functions were used to estimate the risk of crop-loss in terms of the Conditional Probability of Non-Exceedance (CPNE) given by eq. 7 and 8, where $Y_{th-dr}$ is the drought threshold, respectively -0.84 and 40 when the SPEI and VHI/TCI are used.

$$CPNE_{-X_{std}|drought} = Pr(X_{sim} \leq -X_{std}|Y_{sim} \leq Y_{th-dr}) \tag{7}$$

$$CPNE_{-X_{std}|non\_drought} = Pr(X_{sim} \leq -X_{std}|Y_{sim} > Y_{th-dr}) \tag{8}$$

For the purpose of validation and estimation of confidence intervals, the theoretical values of the above CPNE were inferred from the copula functions using the Eq. 9 and Eq. 10 (deduced from the definition of conditional probability),

$$CPNE_{-X_{std}|drought} = \frac{C(u_{-std}, v_{th-dr})}{v_{th-dr}} \tag{9}$$

$$CPNE_{-X_{std}|non\_drought} = \frac{u_{-std} - C(u_{-std}, v_{th-dr})}{1 - v_{th-dr}} \tag{10}$$

where $u_{-std} = F_X(-X_{std})$ and $v_{th-dr} = F_Y(Y_{th-dr})$ are the marginal probabilities of crop-loss and drought occurrence obtained from the kernel-based univariate CDFs. The lower and upper bound of the 95% confidence interval (ci) of the estimated copula dependence parameters were considered using the Eq. 9 and 10 in order to obtain the confidence interval of CPNE coming from the inaccuracy of the copula parameter and to address if the CPNE using simulations (Eq. 7 and Eq. 8) lies within the 95% confidence level.

In sum, first we describe the joint probability of drought hazard and yield anomalies and simulate pairs of data preserving their dependence structure. After that, probability of crop-loss (PNE) and conditional probability of crop-loss (CPNE) are estimated,





addressing whether the probability of crop-loss under drought conditions is higher than during non-drought conditions, and if distinguish drought severity is important. The probability distributions (based on a normal kernel function) of the generated yield anomalies are also analysed for graphical visualisation of the area corresponding to crop-loss.

## 3 Results and discussion

### 3.1 Fitting copula models

The non-parametric estimations of the CDF using a kernel estimator suggest a good agreement with the ECDF (Fig. 3) and therefore, were used to transform the variables to the unit scale (pseudo-observations) for the copula modelling. Figure 3 shows the CDFs and the scatters of the transformed variables and illustrates the crop-loss and drought thresholds used in this study ($-X_{std}$ and $Y_{th-dr}$ respectively). A straightforward way of visualization of the association between the cereal yields and

drought conditions was first carried out based on the scattering of the uniform pseudo-observations of the margins (Fig. 3 – bottom). Most of the transformed data points are concentrated along the diagonal line (Fig. 3 – bottom), mainly due to the correlations between the yield and selected drought indicators (Ribeiro et al., 2018). Most of the works based on copulas have estimates of the marginal distribution functions (Afshar et al., 2016; Bokusheva et al., 2016; Mirabbasi et al., 2012), whereas this procedure has no requirement for prior knowledge of the marginal distributions, entailing therefore less heavy assumptions.

The estimates of the dependence between the yield anomalies and drought indicators were performed using the copula functions from Table 2 (Gaussian, t-copula, Clayton, Frank and Gumbel). Table 3 indicates each copula dependence parameter estimates (ρ, df or θ) and respective AIC values. Based on the values of AIC, a Gaussian copula, a Clayton copula and two Gumbel copulas were eligible to perform the best fits (Table 3). In general, the Archimedean copulas are better suited to estimate the joint distributions between crop yield and drought indicators in most of the cases (Table 3), with the exception of

barley in cluster 1, which is better fitted by a Gaussian copula. Given that AIC penalizes the number of estimated parameters (Wilks, 2006), t-copulas are not expected to be chosen, since they have two parameters that control the tail dependence.

The selected copula functions (Table 3) suggest that, in general, the relationship between yield and drought conditions is described by an asymmetric dependence in the tails of the joint distributions, except in the case of barley in cluster 1. This feature is illustrated in Fig. 4, showing the different shapes and contours of the selected copula densities. While wheat in cluster

1 and 2 shows a stronger dependence in the upper tail of the joint distributions based on Gumbel copulas (suggesting higher probability of observing a higher value of yield anomalies given a high value of the drought indicators), barley in cluster 2 shows stronger dependence in the lower-left tail based on a Clayton copula, suggesting higher probability of observing a lower value of yield anomalies given a low value of the drought indicators. The randomly generated yield and drought data was transformed back to the original scales (Fig. 4 bottom panel) and the respective scatter plots indicate that more extreme values

are generated using the joint distribution relationships. In general, the modelling of the joint distributions leads to results close to the real observations (Fig. 4 bottom panel).





In the present study we cannot conclude about the adequacy of the copula models to a specific type of drought indicator (remote sensing or hydro-meteorological), since only one type of drought indicator was considered for each cereal. In contrast, Bokusheva et al. (2016) have found that Gumbel copulas provided better fits representing the joint distribution of VCI and wheat, while Frank copulas described better the dependence between TCI and wheat yields, in Kazakhstan. Madadgar (2017)

modelled the conditional probability density functions of crop yields under wet and dry conditions using SPI and SSI and found that a Clayton copula was the best function to model the dependence structures. However, the referred studies were somehow more restrictive as they do not take advantage from using both remote sensing and hydro-meteorological drought indicators, and do not select the most important one *a priori*.

### 3.2 Probability of Non-Exceedance and Conditional Probability of Non-Exceedance using copula simulations

After estimating the joint distribution functions and simulating pairs of data preserving the modelled dependence structures, we evaluate and compare the Probability of Non-Exceedance (PNE) and Conditional Probability of Non-Exceedance (CPNE) as a function of the crop-loss threshold. In this way, we evaluate if distinguishing drought severity leads to different risk values of crop-loss in comparison to disregarding a drought threshold (using only simulations of yield) and compare the probability of crop-loss under drought and non-drought conditions (using both simulations of yield and respective drought indicator). One

of the key advantages of estimating the values of PNE and CPNE by means of the copula simulations is the use of larger samples which entail more joint extreme values based on the joint behaviour of crop yields and drought hazard.

Figure 5 shows the PNE curves and the distributions of the simulations of yield anomalies with the respective crop-loss area correspondent to the probability (%) of the yield anomaly not exceeding -1std. The PNE curves indicate more than 19% chance of having crop losses in all cases. According to Fig. 5, wheat at cluster 1 is the cereal with the highest risk level (22%) followed

by barley in cluster 1 (19.8%), wheat in cluster 2 (19.4%) and barley in cluster 2 (19.2%) (Fig. 5). As a matter of fact, the wheat's left tail area (negative yield anomalies) is slightly higher in cluster 1, suggesting a higher risk of wheat loss in the northern sector of the IP.

The following target was to compare the likelihood of crop-loss under drought and non-drought conditions. Figure 6 shows the simulated crop yield anomalies during drought (red left-sided boxplots) and non-drought (blue right-sided boxplots) events.

As expected, the boxplots show lower (and negative in average) values of yield anomalies during drought events in comparison with non-drought episodes. Although the number of samples simulated under drought conditions is smaller than under non-drought conditions (Fig 6), the use of copula simulations enhances the amount of simulated joint low extremes (i.e. co-occurrence of crop-loss and drought events).

The differences in terms of crop-losses between cereals and regions is much evident when differentiating the climatic

conditions (Fig. 7), particularly during drought conditions. Figure 7 shows that the values of CPNE under drought (non-drought) conditions are above (below) the values of PNE illustrated in Fig. 5. As a matter of fact, the distributions of the yield simulations during drought events show a shift to the left towards positive values of yield anomalies, while the distributions of yield simulations during non-drought events show a shift to the right towards positive values of yield anomalies (Fig. 7). A



major conclusion from Fig. 7 is that crop anomalies decline much more during drought years, while fewer losses are expected during non-drought years, suggesting a high agricultural drought risk level of wheat and barley in both clusters. While in Fig. 5 the values of PNE the crop-loss threshold range between 22% (wheat in cluster 1) and 19.2% (barley in cluster 2), the values of CPNE the crop-loss threshold during drought years range from 59.2% (barley in cluster 1) to 36.5% (wheat in cluster 1).

The higher probability of crop-loss obtained when analysing only drought years agrees with Páscoa et al. (2017b), which have shown a very high agreement between low wheat yield anomalies and drought conditions in the IP, even on provinces where the linear correlation is no-significant.

The case of barley in cluster 1 during drought conditions is quite distinct: the distribution of yield simulations is steeper in cluster 1, and flatter in cluster 2 (Fig. 7). Moreover, the distribution of barley in cluster 1 is more shifted to negative yield

anomalies, stressing that the risk of barley is higher on cluster 1 (59.2%) than on cluster 2 (39.4%) when considering only drought years (while it is quite similar on both clusters in Fig. 5). The risk level of barley-loss in cluster 1 during drought years is almost 3 times higher than the value illustrated in Fig. 5 (19.8%), supporting the importance of conditional probabilities for agricultural drought risk purposes. The conditional probability of wheat-loss (Fig. 7), is also higher when considering drought years only, although it is less than two times higher than the values obtained with all climate conditions (Fig. 5). Among all

the cases, the wheat in cluster 2 is the second highest case of conditional probability of crop-loss under dry conditions (46.7%). While barley suggests higher conditional probabilities of crop-loss under drought conditions in cluster 1, wheat suggests higher conditional probabilities of crop-loss under drought conditions in cluster 2.

Although there is a greater risk of crop-loss during drought years (Fig. 7), some losses can still be expected during non-drought years, particularly in cluster 2 (14.1% and 7.77% in the case of wheat and barley, respectively). In the northern sector (cluster

1) the probabilities of crop-loss under non-drought conditions have the lower values, displaying 3.97% in the case of wheat and 3.65% in the case of barley. Some studies point to crop damages attributable to excessive wet soils (Rosenzweig et al., 2002; Zampieri et al., 2017) due to delayed planting or later harvest, nutrients runoff, development of pests and diseases, among others, highlighting the complexity of quantifying agricultural risk levels for management purposes, and the non-linear relation between crop yield and climate conditions. The lower values of CPNE under non-drought conditions in cluster 1

support the fact the slightly high values of PNE in cluster 1 illustrated in Fig. 5 are mainly dominated by drought conditions.

The theoretical CPNE based on Eq. 9 and 10 (Table 4) agrees quite well with the estimates of the CPNE in Fig. 7, thus corroborating the representativeness of the copula experiment using 1000 simulations. Nevertheless, the use of simulations allows to increase the sample size and to generate more joint extreme values based on the dependence structures characterized by the selected copulas. In addition, the effect of the copula parameters (ρ or θ) inaccuracy due to the finiteness of available

sample is considered in Table 4 in terms of the 95% confidence level interval of CPNE based on the confidence interval of the copula parameters taken from Table 3. Table 4 shows that the theoretical CPNE under drought conditions still remain well above the CPNE under non-drought, with their difference taking the smallest value at the lower bound of the copula parameter confidence interval. In most cases, those differences are positive, as expected from the effect of drought on crop yield, despite the relative finiteness of the sample to fit the copula models.





The results show that CPNE based on simulations (Fig. 7) and theoretical equations (Table 4) indicate greater probabilities of crop-loss under drought conditions in most of the cases, even considering the two-sided confidence bound values of the copula parameter. Moreover, the results indicate that the CPNE using the simulations (Fig. 7) lies within the estimates of CPNE using the two-sided confidence bound values of the copula parameter at the 95% level of confidence (Table 4). The only exception

is the case of barley in cluster 2 considering the lower bound of θ, which gives greater probabilities of crop-loss during non-drought conditions rather than during drought conditions, suggesting that other factor than water stress is the cause of crop-failure. This result has to do with the negative value of the copula parameter in the lower confidence bound (θ = -0.38), thus suggesting a weak dependence between crop-loss and drought conditions in this case. However, at the 80% confidence level (θ∈[0.03, 1.55]) the values of the copula parameter confidence bounds are both positive and give higher CPNE under drought

conditions. This lack of accuracy of the CPNE at the 95% in the case of barley in cluster 2 may be the reason why the CPNE under drought conditions are not the highest of all cases, as would be expected from a Clayton copula (which is known for capturing lower tail dependence).

The overall results show the importance of the concept of conditional probability to distinguish different meteorological settings and the applicability of the Copula Theory to analyse joint extreme values. The use of copula simulations for the

analysis of the co-occurrence of dry and low-yield extreme events have shown the additional value of this methodology for the accurate estimation of drought-related crop-failure. Moreover, this study suggests the relevance of impact-centric approaches (also referred in literature as 'bottom-up' approaches (Zscheischler et al., 2018)) to identify and characterize the hazards which lead to the larger impacts.

## 4 Conclusions

This study investigated the usefulness of copula methods in estimating the likelihood of drought risk in wheat and barley cropping systems, when applied to two regions in IP. Here we proposed to model the joint probability of yield and drought hazard using copulas, based on a prior analysis of association between drought and crop-loss (Ribeiro et al., 2018). The advantage of using a probabilistic approach is to meet the ambitious challenge of helping farmers and stakeholders in managing their operations, by identifying the probability of crop-loss under specific drought conditions. Hernández-Barrera and

Rodríguez-Puebla (2017) and Ferrise et al. (2011) have shown that projected warmer and drier climate will lead to wheat yield shortfall over the IP and Mediterranean, respectively, highlighting the importance of establishing novel statistical approaches for agricultural drought risk analysis. Other crops rather than rainfed cereals are also expecting significant losses during the next century in the IP (Quiroga and Iglesias, 2009; Resco et al., 2010; Saadi et al., 2015), and the here proposed crop-specific approach could be applied to other agricultural systems under drought conditions for different regions around the world.

The novelty of the presented models, in comparison to other works addressing climate risk in the IP (*e.g.* Iglesias and Quiroga, 2007; Resco et al., 2010; Ribeiro et al., 2018b), is the focus on the impacts associated with droughts and on the joint probability of rainfed yield anomalies and drought hazard. Previous works using copulas in hydro-climatology studies have tended to



focus on the joint distribution of different characteristics of the hazard events, such as frequency, intensity, severity, duration, among others (Chen et al., 2013; Li et al., 2015; Mirabbasi et al., 2012). Moreover, the restriction to the bivariate case allowed for a simpler interpretation of the results, in contrast to higher dimension copulas (Ganguli and Reddy 2013; Afshar et al. 2016). More recently, copulas have been applied to estimate the joint behaviour of drought conditions and the associated

impacts in agricultural systems (Bokusheva et al. 2016; Madadgar et al. 2017), instead of using drought information only. We have adopted a similar approach to reproduce time-, regional-, and crop-specific dependence of drought conditions, and the probability distribution of crop yield anomalies under drought conditions was estimated for risk analysis. In addition, the use of different drought indicators in this study represents an advantage since crops react differently to several factors at distinct moments and locations, highlighting the importance of quantifying the contributions of different drought indices on a regional

scale (Zarei et al., 2013).

Overall, the results of the estimated copula functions have shown that the dependence structure between crop yield anomalies and drought conditions is mainly asymmetrical (Fig. 3), suggesting a dependence between extreme values of rainfed cereal yield anomalies and drought indicators, and the subsequent simulated distributions of crop yield anomalies are quite consistent with the observations. The results highlighted that the use of copulas for probabilistic assessment allow the estimation of the

dependence in the tails of the distribution and were able to give the likelihood of crop-loss under drought conditions. This feature is of the most interest in risk analysis given that it models the joint probability of occurrence of crop-loss and drier events. In summary, we have shown the existence of dependence among extreme values of yield anomalies and drought indicators, and that yield anomalies decline much more during drought years rather than during non-drought years. The agricultural drought risk levels estimated in the present work aim to improve the effectiveness of the agricultural management

of rainfed cereals in the major agricultural areas of the IP.

To further the research, the calculation of SPEI using climate projections of precipitation and temperature holds an added-value to estimate drought risk levels for the next century. Likewise, the use of seasonal drought forecasts is also quite plausible in the approach presented in this study. Nevertheless, the presented results indicated the likelihood of crop-loss based on drought conditions observed much earlier than the harvest time, particularly in cluster 2 using SPEI (February and April with

5 and 1 month of time-scale). Hence, given the uncertainty associated to the seasonal forecasts for regional drought predictability in the IP, the use of past information for predictability studies is still successfully used (Ribeiro and Pires 2016) and is a source of information from an operational point of view. Other potential usefulness of this methodology for future research is the evaluation of its suitability at the province level of the IP territory.

### Acknowledgements

This work was partially supported by national funds through FCT (Fundação para a Ciência e a Tecnologia, Portugal) under projects CLMALERT (ERA4CS/0005/2016), IMDROFLOOD (WaterJPI/0004/2014) and IMPECAF (PTDC/CTA-CLI/28902/2017). Ana Russo and Andreia Ribeiro thank FCT for grants SFRH/BPD/99757/2014 and PD/BD/114481/2016,





respectively. The authors are also sincerely thankful to Ricardo Trigo (Instituto Dom Luiz) for his valuable suggestions and support.

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



**Table 1 - Variables used for copula application. In the first column, the numbers 1 and 2 correspond to the respective provincial cluster (clusters 1 and 2). In the second column, the numbers correspond to the selected weeks in the case of the remote sensing indices, and to the selected months and time-scales (in months) in the case of SPEI. The values of the standardized regression coefficients were determined by Ribeiro et al. (2018).**

| Cereal (X) | Drought indicator (Y) | Standardized regression coefficients |
|---|---|---|
| Wheat 1 | TCI 23 | 0.76 |
| Barley 1 | VHI 22 | 0.91 |
| Wheat 2 | SPEI 4-1 | 1.05 |
| Barley 2 | SPEI 2-5 | 1.07 |

**Table 2 - Equations of the copula functions, where $u$ and $v$ are univariate variables, $\emptyset^{-1}$ is the inverse of standard Gaussian CDF, $t_{df}^{-1}$ is the inverse t–student CDF, $df$ is the degree of freedom, $\rho$ and $\theta$ are dependence parameters.**

| Family | Joint Cumulative Distribution Function $C(u,v)$ | Parameter range |
|---|---|---|
| Gaussian | $\int_{-\infty}^{\emptyset^{-1}(u)} \int_{-\infty}^{\emptyset^{-1}(v)} \frac{1}{2\pi\sqrt{(1-\rho^2)}} e\left(-\frac{u^2+v^2-2\rho uv}{2(1-\rho^2)}\right) du dv$ | $-1 \leq \rho \leq 1$ |
| t | $\int_{-\infty}^{t_{df}^{-1}(u)} \int_{-\infty}^{t_{df}^{-1}(v)} \frac{1}{2\pi\sqrt{(1-\rho^2)}} e\left(1+\frac{u^2+v^2-2\rho uv}{df(1-\rho^2)}\right)^{-\frac{df+2}{2}} du dv$ | $-1 \leq \rho \leq 1$ $df \geq 1$ |
| Clayton | $(u^{-\theta}+v^{-\theta}-1)^{\frac{1}{\theta}}$ | $\in [-1,\infty[/\{0\}$ |
| Frank | $-\frac{1}{\theta}\ln\left(1+\frac{(e^{-\theta u}-1)(e^{-\theta v}-1)}{e^{-\theta}-1}\right)$ | $\theta \neq 0$ |
| Gumbel | $e^{-[(-\ln u)^\theta+(-\ln v)^\theta]^{\frac{1}{\theta}}}$ | $|\theta| < \infty$ |

10   **Table 3 - Copula dependence parameter estimates ($\rho$,df or $\theta$), 95% confidence interval (ci) in parenthesis and AIC values. The ci 95% denoted by '-' indicates that the model was unable to compute the ci using the profile likelihood of the parameter. The selected models according to the lowest value of AIC are in bold.**

|  | Gaussian | | | t-copula | | | Clayton | | | Frank | | | Gumbel | | |
|---|---|---|---|---|---|---|---|---|---|---|---|---|---|---|---|
|  | ρ | ci 95% | AIC | df | ci 95% | AIC | θ | ci 95% | AIC | θ | ci 95% | AIC | θ | ci 95% | AIC |
| **W1** | 0.63 | 0.33;0.82 | -11.79 | 0.75 | 0.02;2.23 | -3.73 | 1.91 | 1.14,2.68 | -11.07 | 6.45 | 3.95,8.95 | -13.42 | **2.34** | **1.72,2.96** | **-16.7** |
| **B1** | **0.88** | **0.80;0.96** | **-39.1** | 0.92 | - | 4 | 4.09 | 2.21,5.96 | -36.97 | 12.5 | 4.90,20.1 | -38.93 | 3.11 | 1.89,4.33 | -32.43 |
| **W2** | 0.54 | 0.15;0.74 | -7.23 | 0.54 | -0.25;2.82 | -3.55 | 1.35 | 0.56,2.13 | -7.95 | 4.35 | 1.83,6.88 | -6.69 | **1.81** | **1.24,2.38** | **-8.78** |
| **B2** | 0.32 | -0.07;0.62 | -0.99 | 0.42 | -14.02;21.96 | 3.02 | **0.79** | **-0.38,1.95** | **-2.70** | 2.54 | -0.06,5.14 | -1.06 | 1.42 | 0.96,1.88 | -1.12 |





**Table 4 – Theoretical CPNE (%) during drought and non-drought conditions (Eq. 9 and 10) and respective lower and upper bounds of the 95% confidence interval, where $u_{-std}$ and $v_{th-dr}$ are the marginal probabilities of crop-loss and drought occurrence, and θ or ρ are the estimated copula parameters with 95% confidence limits (Table 3). The only exception which gives greater values of CPNE during non-drought conditions rather than drought is denoted by '*'.**

| | Copula | $u_{-std}$ | $v_{th-dr}$ | θ or ρ | CPNE drought | CPNE non-drought | Lower confidence bound (95%) | | | Upper confidence bound (95%) | | |
|---|---|---|---|---|---|---|---|---|---|---|---|---|
| | | | | | | | θ or ρ | CPNE drought | CPNE non-drought | θ or ρ | CPNE drought | CPNE non-drought |
| **W1** | Gumbel | 0.22 | 0.51 | 2.34 | 39.3 | 4.00 | 1.72 | 35.4 | 8.44 | 2.96 | 41.2 | 1.98 |
| **B1** | Gaussian | 0.20 | 0.27 | 0.88 | 62.4 | 4.31 | 0.80 | 56.3 | 6.54 | 0.96 | 70.3 | 1.40 |
| **W2** | Gumbel | 0.19 | 0.25 | 1.81 | 42.5 | 11.2 | 1.24 | 27.8 | 16.1 | 2.38 | 51.5 | 8.17 |
| **B2** | Clayton | 0.19 | 0.29 | 0.79 | 41.1 | 9.99 | -0.38 | 6.55* | 24.1* | 1.95 | 55.2 | 4.23 |

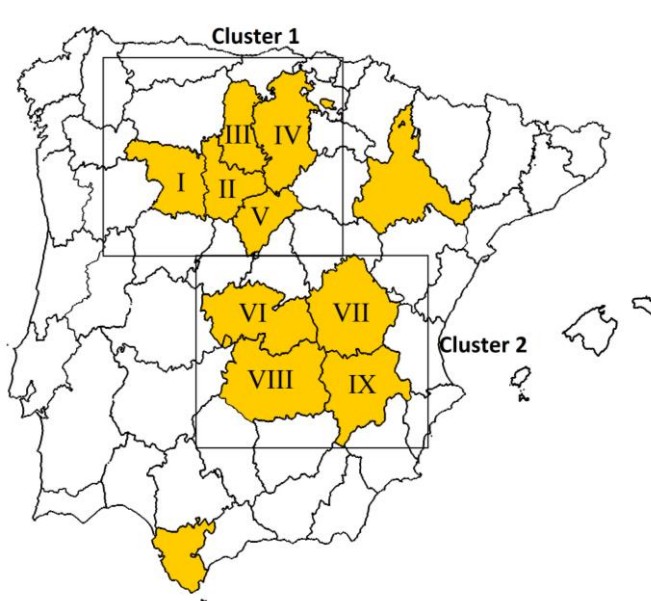

10 **Figure 1: Provinces with more than 50% of the territory occupied by agricultural areas and more than 50% of rainfed crops (yellow) according to CLC2012, and selected clusters of provinces. Cluster 1 provinces: Zamora (I), Valladolid (II), Palencia (III), Burgos (IV) and Segovia (V). Cluster 2 provinces: Toledo (VI), Cuenca (VII), Ciudad Real (VIII) and Albacete (IX).**



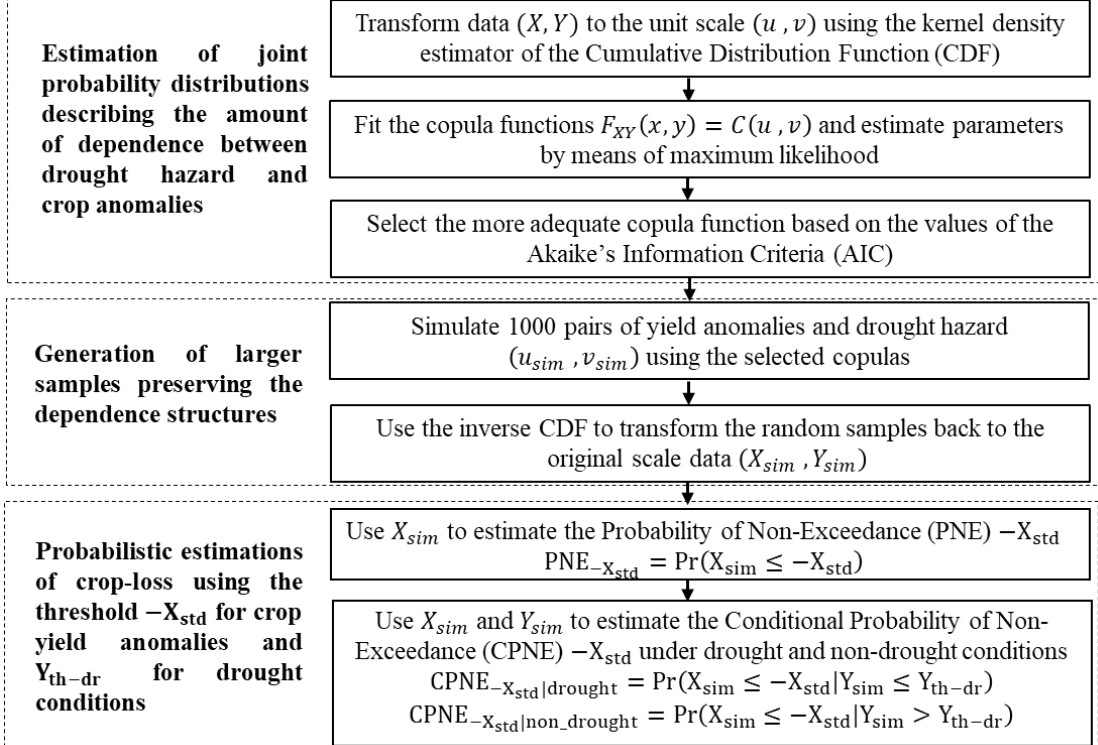

**Estimation of joint probability distributions describing the amount of dependence between drought hazard and crop anomalies**

Transform data $(X, Y)$ to the unit scale $(u, v)$ using the kernel density estimator of the Cumulative Distribution Function (CDF)

Fit the copula functions $F_{XY}(x, y) = C(u, v)$ and estimate parameters by means of maximum likelihood

Select the more adequate copula function based on the values of the Akaike's Information Criteria (AIC)

**Generation of larger samples preserving the dependence structures**

Simulate 1000 pairs of yield anomalies and drought hazard $(u_{sim}, v_{sim})$ using the selected copulas

Use the inverse CDF to transform the random samples back to the original scale data $(X_{sim}, Y_{sim})$

**Probabilistic estimations of crop-loss using the threshold $-X_{std}$ for crop yield anomalies and $Y_{th-dr}$ for drought conditions**

Use $X_{sim}$ to estimate the Probability of Non-Exceedance (PNE) $-X_{std}$
$$PNE_{-X_{std}} = \Pr(X_{sim} \leq -X_{std})$$

Use $X_{sim}$ and $Y_{sim}$ to estimate the Conditional Probability of Non-Exceedance (CPNE) $-X_{std}$ under drought and non-drought conditions
$$CPNE_{-X_{std}|drought} = \Pr(X_{sim} \leq -X_{std}|Y_{sim} \leq Y_{th-dr})$$
$$CPNE_{-X_{std}|non\_drought} = \Pr(X_{sim} \leq -X_{std}|Y_{sim} > Y_{th-dr})$$

**Figure 2 – Scheme of the copula-based approach adopted in the present study.**



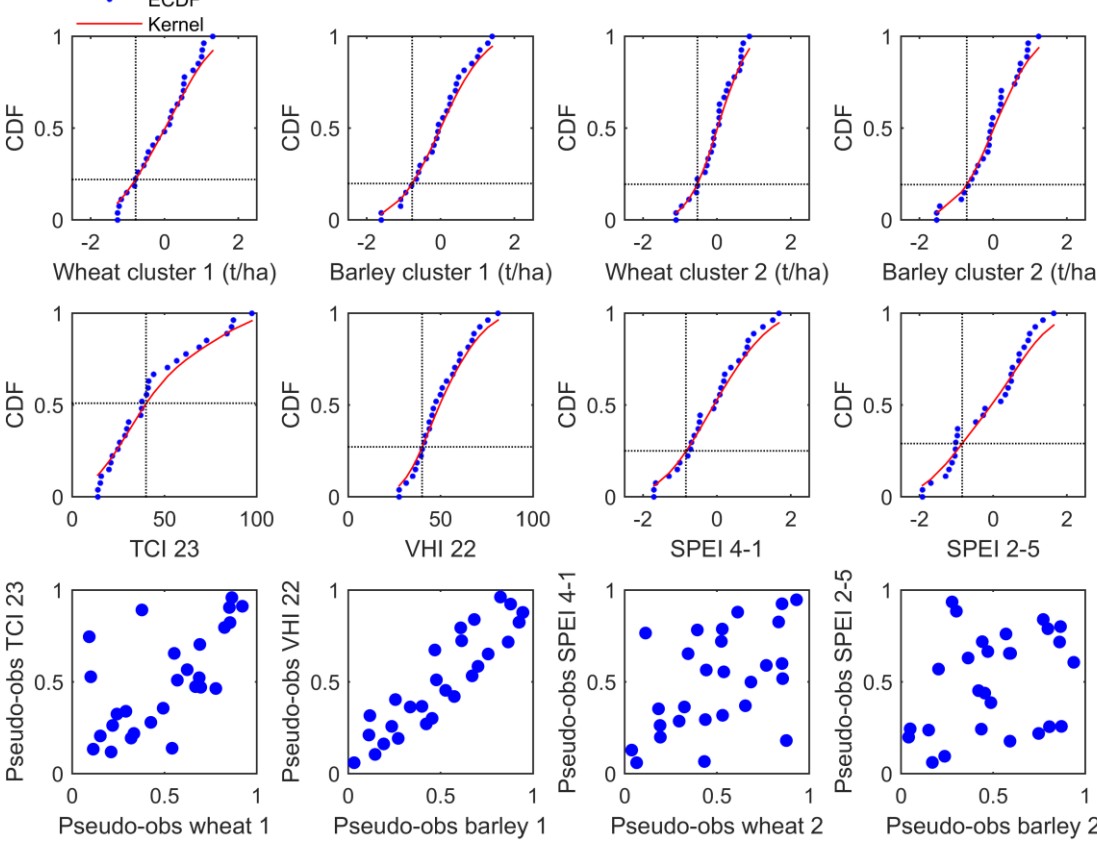

**Figure 3 – Empirical cumulative distribution functions (ECDF, blue points), kernel density estimation of the CDF (red line), crop-loss and drought thresholds (dotted black vertical line) and pseudo-observations (scatter) of the margins on the interval [0,1].**



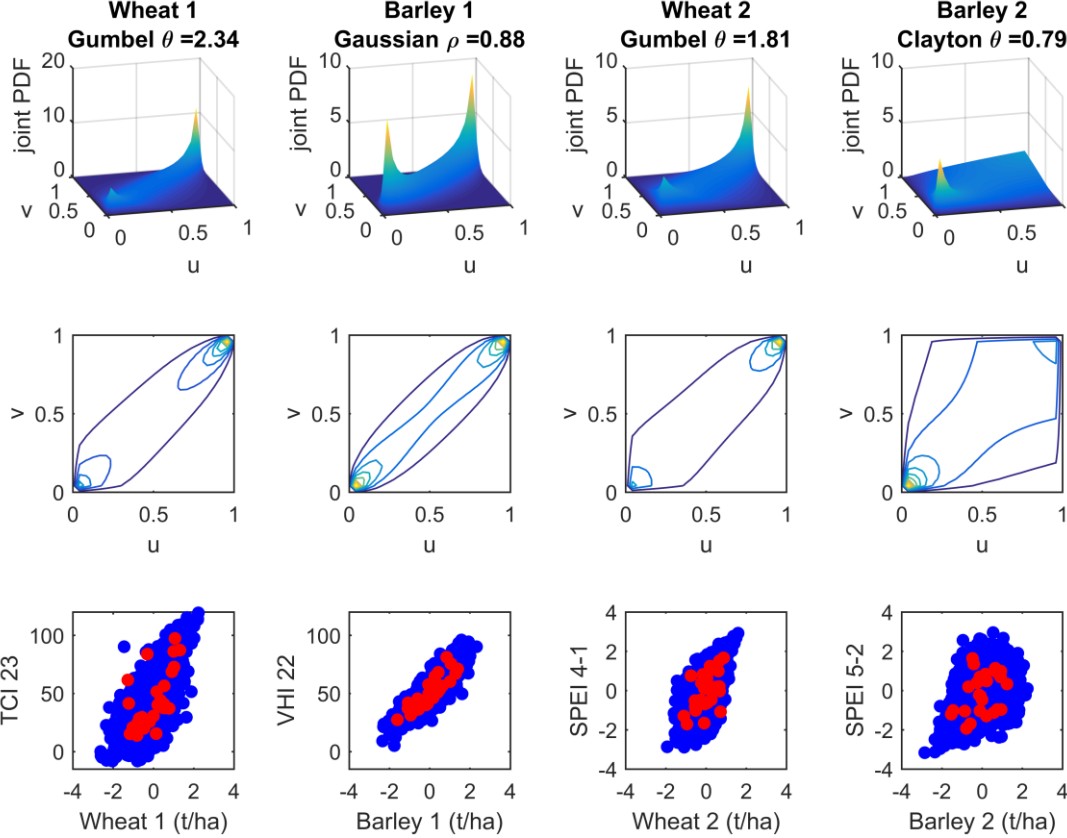

**Figure 4 – Selected joint Probability Distribution Functions (PDF) where *u* and *v* are scalar values on the interval [0,1] (top), contours showing the two-dimensional view of PDFs (middle) and observed (red) and copula-based simulations (blue) scatter plots of crop yields and drought indicators (bottom).**




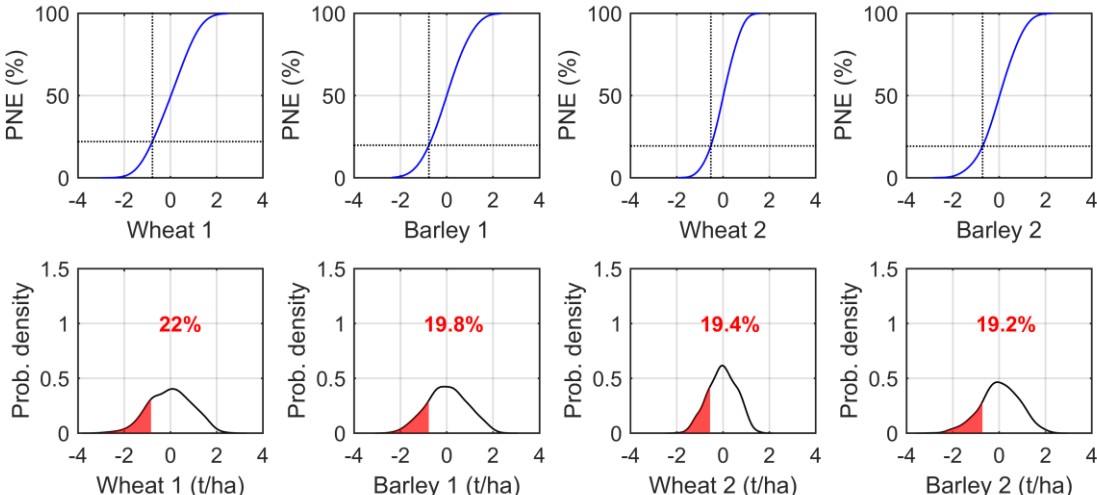

**Figure 5 – Probability of non-exceedance (PNE) function (%) of yield anomalies (top) in both clusters based on the derived simulations from the estimated copulas and respective probability density estimates (bottom). On the bottom panels, the red values indicate the probability of crop-loss which is also indicated in the top panels by the intersected dashed lines indicating the threshold of crop-loss and respective PNE value.**

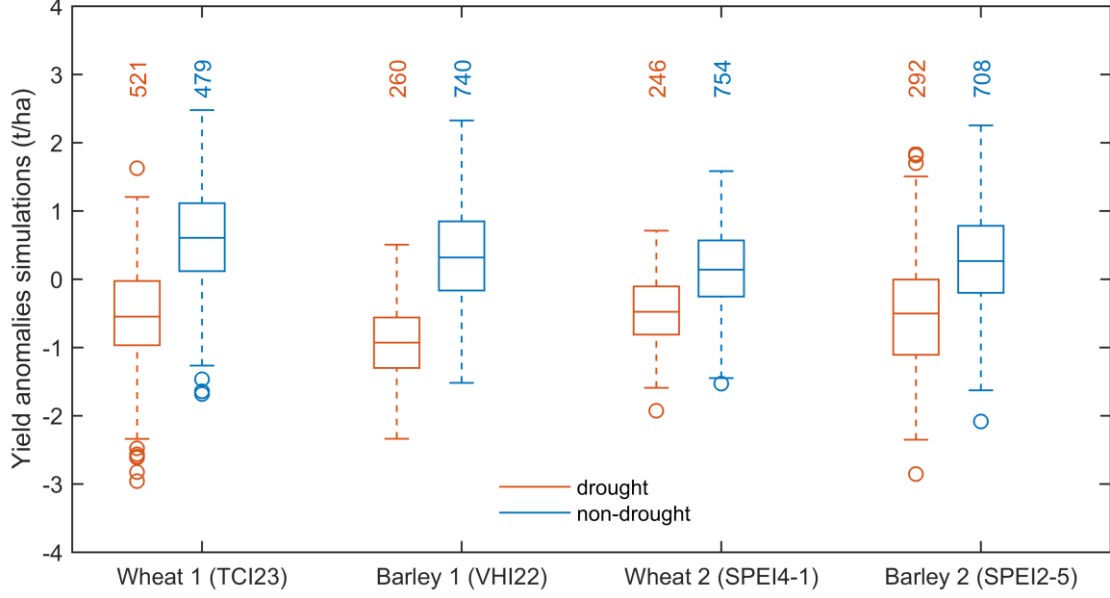

**Figure 6 - Wheat and barley yield simulations differentiating drought (red) and non-drought conditions (blue) according to the respective drought indicator denoted in parenthesis in the x-tick label. The numbers on top of the boxplots denote the sample size of the simulations under the different climatic conditions.**



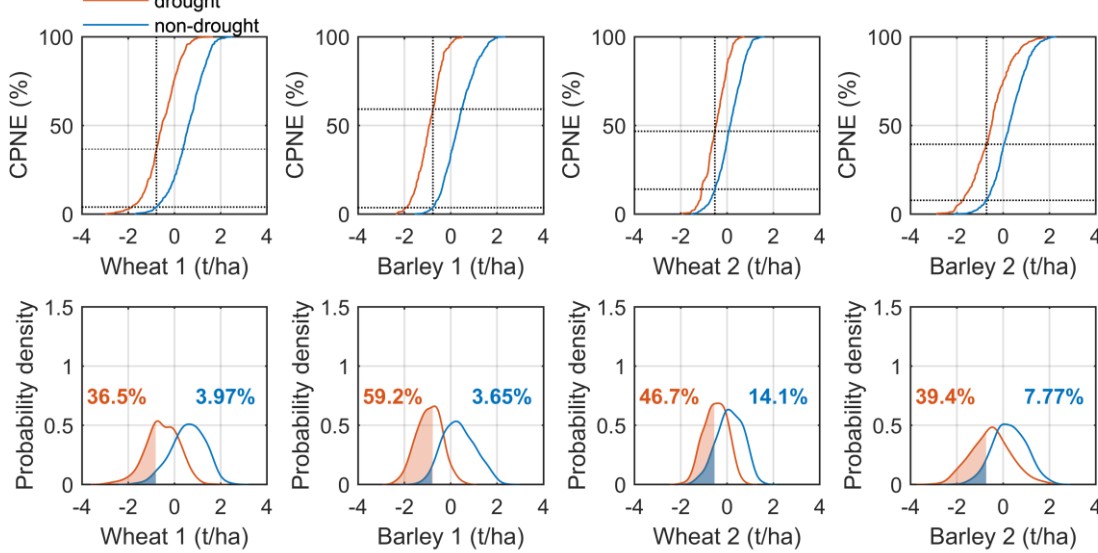

**Figure 7 – Conditional probability of non-exceedance (CPNE) function (%) based on the derived copula simulations (top) and respective probability density estimates (bottom) under drought (orange) and non-drought conditions (blue). On the bottom panels, the orange and blue values indicate the probability of crop-loss under the different climatic conditions, which is also indicated in the top panels by the intersected dashed lines indicating the threshold of crop-loss and respective CPNE value.**