# Peer review of "Probabilistic modelling of the dependence between rainfed crops and drought hazard"

_Natural Hazards and Earth System Sciences, 2019_

## Referee Comment (RC1) · Anonymous Referee #1 · 6 May 2019

The study aims at advancing agricultural drought risk management through providing a probabilistic model for assessing the risk of crop loss and drought for two regions of the Iberian Peninsula with rainfed cropping systems. The authors apply the concept of copula models for this purpose and infer probabilities of crop loss under drought and non-drought conditions for the two regions and two crops.

Overall, the authors address an important topic by transferring the use of copula models to agricultural drought risk management. The paper is very well organized, easy to follow, and clearly written. The methods are precisely described, including a flowchart of the concept. Also, adequate reference to existing literature is presented.

However, I am concerned about two issues that should be further addressed:

[Figure]

1) Data and methods: The study focuses on crop loss of rainfed agriculture due to drought. While the data base is 27 years, only 7 years seem to be years with drought conditions according to Figure 3. Since only one value per year is retained for analysis, this seems like a very small sample size for making robust probabilistic inferences, especially regarding the tail of the distribution. One could argue that the study is not only focused on drought, since crop loss under non-drought conditions is also investigated; however, if the authors' intention was to assess the risk of crop loss more generally, in my view a stronger focus on other hazards or drivers of crop loss would have been beneficial to include. In addition to the sample size issue, the spatial and temporal scale of the selected drought indicators seems rather coarse. Spatial scale: I assume some spatial average or aggregate measure of drought indicators over the two clusters was used (not clear from the manuscript). How representative is such a spatially aggregated drought intensity measure for explaining crop loss, which likely results from small-scale interactions of hazard, vulnerability of the plants, soil type, additional irrigation systems in parts of the clusters, etc.? Temporal scale: monthly SPEI (as used for cluster two) is a commonly used indicator; however, for this specific purpose, wouldn't sub-monthly drought conditions be more suitable given the effect during different growth stages, as the authors point out themselves? Given these data issues, I am a bit concerned about the meaningfulness of the inferred results on probability of crop loss under drought and non-drought, as presented in section 3.2.

2) Aim of the study versus methodological setup: The aim of the study is to quantify the risk of crop-loss under drought and non-drought conditions for two regions and two crop types; for this purpose the authors model the dependence between drought conditions and crop anomalies. What actually is the practical value of assessing the probability of rainfed crop loss under non-drought conditions with this setup, where the focus is solely on drought indicators? The aim of the study one the one hand and the deployed data and statistical methods one the other hand seem a bit separated or not well aligned at times. While for drought conditions the data does not seem fully suitable for analysis (see issue 1 on sample size), for non-drought conditions the value of the

analysis is not entirely clear to me, at least how it is presented currently. It would be beneficial if the authors could elaborate further on the applicability of their method for agricultural drought risk management.

Detailed comments:

2.1 A bit more background how the drought indicators per cluster and crop were calculated and selected in the authors' previous study would be useful (especially how the spatially variable indicators were regionally averaged). In contrast, details about SPEI and vegetation condition indicator calculation could be shortened by pointing to existing references.

Fig. 3: What is the dotted horizontal line?

Fig. 4 bottom row: Due to the filled marker style it is hard to discern details; I'd suggest making a density plot of points.

References: duplicate reference Ribeiro 2018a and b

---

## Referee Comment (RC2) · Anonymous Referee #2 · 12 May 2019

General comments: The topic of this study is highly relevant, as the Mediterranean area is in need of strategies to cope with weather-related risks, especially drought. The effort to characterize the drought –crop yield losses are therefore valuable. However, in my view, there is a major issue in this manuscript that should be addressed before publication. This is the lack of generalized conclusions. Analyses are sound but the time and spatial extent and resolution of the data make difficult to extract general conclusions (I agree with Referee 1 on this issue). Even though, the paper would improve a lot if the authors focus on the findings to characterize risk related to drought on these Mediterranean conditions. Findings should be clearly formulated here in a way that the reader can understand what id the added value of the results, beyond the descriptive analysis of the plots at these specific areas.

[Figure]

Specific comments:

- Abstract: the statement "the estimated conditional probabilities suggest that the like-lihood of crop-loss under dry conditions is higher than under non-drought conditions" is pretty obvious for Mediterranean conditions and should not be selected as a main result of the study in the abstract. Please, reformulate to stress the added value of the findings.

- Pag. 2 line 3- This is not recent, It has been lia ke this for decades now, as it is main concern in Mediterranean systems

- Pag. 3 line 6- Referred to which period?

- Pag. 2 lines 14-15- Given that there are many studies using this methodology at the global scale, I would specify here the issues and flaws, but also the advantages, detected by the authors to enrich the discussion.

- Pag. 4 line 6- Somewhere in the paper the low resolution (0.5°) of these data and its implications for analysis accuracy should be discussed and handled. Actually, the resolution used is not clear, as in line 19 4 km is mentioned. Please clarify.

- Pag. 4, line 13- With data from which data soruce? Please justify with Penman-FAO, in principle more accurate methods, is not used.

-Pag. 4 line 29- More details should be provided on these models. This paper should be understandable without reading Ribeiro et al. (2018).

- Pag. 5 lines 2-6- This information is maybe more appropriate for the introduction. The rest of the section until 2.2 is not clear if it is meant as a summary of the data from Ribeiro et al. (2018) or it is new material with a preliminary analysis id data; if this is the case they should be in the result section.

- Section 3.1. More examples for interpreting results in terms of consequences for crop losses-drought weather relationships

- Section3.2. Please see my general comments. Maybe separating between results and discussion sections could help. Alternatively, a paragraph of descriptive results should be followed by an interpretation and then extracting generalized statements when possible. Part of this is done in the conclusions section (see below).

- Conclusions: In my view, these are not really conclusions but a summary of the study or of the discussion. I would recommend addressing the discussion separately, and then to reduce and focus the conclusions section.

---

## Author Comment (AC1) · 7 Jun 2019

We would like to thank the Referee#1 for his/her careful review and constructive feedback, and also for the opportunity to engage in a stimulating discussion. We truly believe that this process will enhance and clarify the paper's content. Author's responses follow below identified as AR next to the Referee#1 comments.

Referee#1: The study aims at advancing agricultural drought risk management through providing a probabilistic model for assessing the risk of crop loss and drought for two regions of the Iberian Peninsula with rainfed cropping systems. The authors apply the concept of copula models for this purpose and infer probabilities of crop loss under drought and non-drought conditions for the two regions and two crops. Overall, the

[Figure]

authors address an important topic by transferring the use of copula models to agricultural drought risk management. The paper is very well organized, easy to follow, and clearly written. The methods are precisely described, including a flowchart of the concept. Also, adequate reference to existing literature is presented. However, I am concerned about two issues that should be further addressed:

Referee#1- 1) Data and methods: The study focuses on crop loss of rainfed agriculture due to drought. While the data base is 27 years, only 7 years seem to be years with drought conditions according to Figure 3. Since only one value per year is retained for analysis, this seems like a very small sample size for making robust probabilistic inferences, especially regarding the tail of the distribution.

AR: In response to the sample size issue, Fig. 3 (2nd row panels) shows that the number of years under drought conditions varies with the location (cluster 1 or 2) and with the type of crop (wheat or barley). For instance, in the case of wheat in cluster 1, from a set of 27 years more than half of the years correspond to drought conditions (14 years). However, in the remaining cases (wheat in cluster 2 and barley in clusters 1 and 2), the number of drought events in the samples are lower, as pointed out by the Referee. This feature highlights the singularity of the extreme events and subsequently how challenging they are. Nevertheless, we totally agree that the ideal situation would be to have more than 27 years of data, however to increase the time period, we would have to sacrifice the use of the remote sensing of vegetation, which is currently an important source of information, complementary to climate variables. We must also stress that in several regions of the world, namely in developing countries, remote sensing data from the last 30 years is the only data that is available for crop losses studies. In addition, we also agree with the Referee that the importance of sample size increases with the complexity of the models and, for this reason, in page 5 lines 24 to 28, we highlight that we have restricted the analysis to the bivariate case using two-dimensional copulas, because the development of higher dimensional copulas exhibits very complex structures and entails additional requirements. Moreover, in page 6 lines

20 to 23 we point out that "The drawback of the shorter sample size is surpassed by the nonparametric estimation of the margins, which avoids heavy assumptions about their distributions, even when the available sample is rather small (Corder and Foreman, 2011; Fahr, 2017)." Nevertheless, despite the fit being performed with n=27, the great asset of the proposed methodology is to estimate probabilities of crop-loss based on 1000 simulations preserving the dependence structure, overcoming the sample size issue in the estimations of the probabilities of crop-loss.

Referee#1: One could argue that the study is not only focused on drought, since crop loss under non-drought conditions is also investigated; however, if the authors' intention was to assess the risk of crop loss more generally, in my view a stronger focus on other hazards or drivers of crop loss would have been beneficial to include.

AR: Thank you for the valuable comment which wisely promotes a deep and stimulating reflection on the nature of the stated problem. In fact, the understanding of the concurrent or sequential occurrence of different hazards which lead to extreme impacts is currently a challenging topic. The combined condition of more than one hazard would be a more complete approach to address agricultural losses in general and the development of higher dimensional models with even more complex structures is an optimal suggestion to expand the scopes of the current study. However, this is a relevant topic that requires a detailed analysis being now out of the scope of the present work. Here we focus on the influence of drought hazard on agricultural productions because drought is recognized as the main cause of crop loss on a global scale (Lesk et al., 2016), with particular relevance in the Mediterranean region (Lesk et al., 2016; Zampieri et al., 2017). The proposed methodology assesses a hazard that is still currently a hot topic in several sectors, including research, and which is likely to be more frequent. In addition, in page 3 line 16 in the introduction section we state the main objective of the present study: "In this study, a copula-based approach is adopted to model the joint probability density function of crop yield and the drought conditions for probabilistic yield assessment" and in the conclusion section (page 11

line 20) we reinforce the idea that "This study investigated the usefulness of copula methods in estimating the likelihood of drought risk in wheat and barley cropping systems, when applied to two regions in IP". Nevertheless, the results of barley in cluster 2 suggest that other factor than water stress is the cause of crop failure (page 11 line 6 and 7), and for this reason the assessment of whether the occurrence of other climate extremes is amplifying the impact of droughts on agricultural productions is the right direction to further next research (however out scope in the present study). We hence propose to improve the future work in the last paragraph (in page 12 line 28) based on this suggestion to: "Other potential usefulness of this methodology for future research is the evaluation of its suitability at the province level and the assessment of whether other hazards (such as heat waves) are amplifying the impact of droughts on crop's harvest."

Referee#1: In addition to the sample size issue, the spatial and temporal scale of the selected drought indicators seems rather coarse. Spatial scale: I assume some spatial average or aggregate measure of drought indicators over the two clusters was used (not clear from the manuscript). How representative is such a spatially aggregated drought intensity measure for explaining crop loss, which likely results from small-scale interactions of hazard, vulnerability of the plants, soil type, additional irrigation systems in parts of the clusters, etc.?

Regarding to the spatial scale issue, we agree that the use of spatial averages over the clusters is not clearly written in the manuscript, and in the below Detailed comments section 2.1 of the Referee#1 we give a more detailed answer with rephrasing suggestions in order to improve this issue in the manuscript. Regarding the representativeness of the used spatial scale, in fact, most of the studies are at the national scale or local case studies. However, despite the small-scale interactions between crops and other factors, as pointed out by the Referee, the availability of fine-scale harvest data with common time periods in order to have a crop-specific database for the Iberian Peninsula is very limited. The use of longer time series at the province level is a good

compromise and explains rather well the response of the cereals to the drought conditions as shown in a previous study by (Páscoa et al., 2017b). Moreover, drought usually affects large areas, and in the Iberian Peninsula, drought events affecting more than 40% of the territory accounted for 33.8% of all the events in the period 1975-2012 (Páscoa et al, 2017a). The spatial averages over the clusters are the result of the exposure analysis to ensure that we are addressing the annual yield response in the regions dominated by rainfed conditions, which are more vulnerable to droughts (Ribeiro et al., 2018).

Referee#1- Temporal scale: monthly SPEI (as used for cluster two) is a commonly used indicator; however, for this specific purpose, wouldn't sub-monthly drought conditions be more suitable given the effect during different growth stages, as the authors point out themselves? Given these data issues, I am a bit concerned about the meaningfulness of the inferred results on probability of crop loss under drought and non-drought, as presented in section 3.2.

Regarding the temporal scale, despite the short-term response of the plant's activity to the dry conditions (here considered by the weekly TCI and VHI) the integrated effect of droughts at the monthly scale or longer (here considered by the monthly SPEI) is also relevant to quantify the mechanisms that can lead to annual crop loss. Moreover, the methodology considered for selection of one drought indicator for each case reflects the importance of the different temporal responses of the cereals to climate conditions along the vegetative cycle and their impact on the cereal's annual yield. In page 4 line 30 to 33, and page 5 line 1 is written in the manuscript: "First, a pool of the drought indicators better related with wheat and barley yield were chosen based on stepwise regression (95% confidence level), selecting the most statistically significant time scales and months of SPEI, together with the weeks of VHI, VCI and TCI. Afterwards, from the pool of selected drought indicators, the one with largest contribution to the yield's variance was selected (Table 1)." Since this aspect is not clear in the article, we propose to rewrite to the following (in page 5 lines 4 to 10): "This selection of

drought indicators highlights that the response of crop yields to climate conditions vary according to the location, type of crop, moment of the vegetative cycle and the temporal scale. While annual yield anomalies in cluster 1 are better characterized by short-term responses to the drought conditions based on the weekly values of TCI and VHI, the annual yield anomalies in cluster 2 are better characterized by the monthly response to the dry conditions based on the SPEI. Moreover, in terms of predictability, the effects of temperature (TCI) and vegetation health (VHI) during late growth stages (weeks 23 and 22 correspond approximately to end of May and beginning of June, respectively for wheat and barley) are the most influential conditions in the northern cluster. On the other hand, the yields in cluster 2 are influenced by drought conditions described by SPEI much earlier, in the beginning of the intermediate growth stages (February and April with 5 and 1 month of time-scale, respectively for wheat and barley). In this way, the importance of including multiple drought response time scales is evidenced for predictability purposes and assessment of drought-related crop-losses." Moreover, to meet the suggestions of Referee#2 regarding this issue, we also propose to move Table 1 and the previous paragraph to the results section.

Referee#1- 2) Aim of the study versus methodological setup: The aim of the study is to quantify the risk of crop-loss under drought and non-drought conditions for two regions and two crop types; for this purpose the authors model the dependence between drought conditions and crop anomalies. What actually is the practical value of assessing the probability of rainfed crop loss under non-drought conditions with this setup, where the focus is solely on drought indicators?

AR: The indicators used in the present study enable the measurement of both dryness and wetness in the case of SPEI, and both stress and good-health of vegetation in the case of VCI and TCI. Hence, the present study is focused on the losses associated to the measures of water balance indicated by SPEI (in the case of cluster 2) and on the losses associated to the measures of vegetation health indicated by VCI and TCI (in the case of cluster 1). As shown by Fig. 6 boxplots, the yield anomalies reduce significantly during drought conditions, while higher yield anomalies values are expected during non-drought conditions (which are characterized by the drought indicators as well). In addition, Fig. 7 shows the distributions of the yield anomalies given different climatic conditions and that droughts lead to major crop losses. A similar setup was considered by Madadgar et al. (2017) in Australia, which assessed droughts based on Standardized Precipitation Index (SPI) and the Standardized Soil-moisture Index (SSI), and have estimated the probability of crop yields exceeding its annual average (while we estimate the probability of non-exceedance -1 standard deviation, i.e., the probability of more extreme crop losses) in dry conditions (SPI or SSI< - 0.5) as compared to normal/wet conditions (SPI or SSI> - 0.5). Similarly, Madadgar et al. (2017) found that a shift from wet (SPI> -0.5) to dry (SPI<-0.5) causes yields of Australian rainfed crops to decrease.

Referee#1: The aim of the study one the one hand and the deployed data and statistical methods one the other hand seem a bit separated or not well aligned at times. While for drought conditions the data does not seem fully suitable for analysis (see issue 1 on sample size), for non-drought conditions the value of the analysis is not entirely clear to me, at least how it is presented currently. It would be beneficial if the authors could elaborate further on the applicability of their method for agricultural drought risk management.

AR: As well stated by the Referee, the fitting is performed with drought years and non-drought years and the probabilities of crop-loss, estimated based on 1000 simulations, distinguish between drought and non-drought conditions. The importance of estimating CPNE|non-drought is to quantify the losses associated to mechanisms not related to dryness directly (for instances flash extreme events during non-drought conditions that lead to crop-loss). In order to discuss whether drought is the main determinant of the crop loss, the following decomposition is required: PNE = (CPNE | drought) * Prob(drought) + (CPNE | no- drought) * Prob (no drought). If drought is the main conditioner then the 1st term is determinant, otherwise it will be the 2nd as it happens

in the case of barley in cluster 2 and almost in the case of wheat in cluster 2. This quantification is important and therefore it is of all relevance to include the probability CPNE | non-drought. In addition, it allows to quantify that the risk of crop-loss increases 32%- 55% (depending on the cereal and cluster) when drought conditions are below the applied severity thresholds. The consideration of other drought severity thresholds for comparison is also a good suggestion to pursue in future agricultural drought risk studies.

Detailed comments: Referee#1: 2.1 A bit more background how the drought indicators per cluster and crop were calculated and selected in the authors' previous study would be useful (especially how the spatially variable indicators were regionally averaged). In contrast, details about SPEI and vegetation condition indicator calculation could be shortened by pointing to existing references.

AR: Thank you for the suggestion, we agree that the use of spatial averages over the clusters is not clearly written, and we propose to rectify that in a revised version. In order to meet this comment from Referee#1 and other suggestion from Referee#2 we propose to remove from Page 4 line 15 the sentence "Further spatial averages were computed for each cluster of provinces." and rephrase in Page 4 line 28 "The spatial averages of VCI, TCI, VHI and SPEI were computed for each provincial cluster and used for further modelling of the joint probability between the drought hazard and cereal yield anomalies. " Moreover, in order to shorten the data section we propose to rewrite from page 4 line 3 to 27 to the following: "Drought conditions were investigated using two types of indices: the hydro-meteorological drought indicator SPEI and the satellite-based Vegetation Condition Index (VCI) (Kogan 1990), the Temperature Condition Index (TCI) (Kogan 1995) and the Vegetation Health Index (VHI) (Kogan 1995). The monthly drought index SPEI gridded values, with spatial resolution of 0.5°, were computed based on precipitation and temperature values from the Climate Research Unit (CRU TS3.21), using a variety of time scales (1 to 12 months). The weekly global maps of VCI, TCI, and VHI were retrieved at 4km spatial resolution from NOAA's ftp server

(ftp://ftp.star.nesdis.noaa.gov/pub/corp/scsb/wguo/data/VHP_4km/geo_TIFF/). While SPEI computation uses climatic water balance anomalies incorporating the role played by the evaporative demand on the occurrence of dry events (Vicente-Serrano et al., 2010), the remote sensing indices characterize the moisture, through the VCI, the temperature induced stress through the TCI and health of vegetation, through the VCI."

Referee#1- Fig. 3: What is the dotted horizontal line?

AR: Thank you for pointing this out. We propose to rewrite the caption of Figure 3 to the following: Figure 3 – Empirical cumulative distribution functions (ECDF, blue points), kernel density estimation of the CDF (red line), crop-loss and drought thresholds (dotted black vertical line), respective marginal probabilities of crop-loss and drought occurrence (dotted black horizontal line), and pseudo-observations (scatter) of the margins on the interval [0,1].

Referee#1- Fig. 4 bottom row: Due to the filled marker style it is hard to discern details; I'd suggest making a density plot of points.

AR: Thank you for the suggestion, we propose the changes presented in the attached Figure and the following changes in the caption: Figure 4 – Selected joint Probability Distribution Functions (PDF) where u and v are scalar values on the interval [0,1] (top), contours showing the two-dimensional view of PDFs (middle) and observed (red triangles) and copula-based simulations (density squares) scatter plots of crop yields and drought indicators (bottom).

Referee#1- References: duplicate reference Ribeiro 2018a and b

AR: Thank you for pointing that out.

References Zampieri et al., Wheat yield loss attributable to heat waves, drought and water excess at global, national and sub-national levels. Environmental Research Letters, 12(6), 064008, https://doi.org/10.1088/1748-9326/aa723b, 2017. Lesk et al., influence of extreme weather disasters on global crop production. Nature, 529,

p.84-87, https://doi.org/10.1038/nature16467, 2016.

Please also note the supplement to this comment:
https://www.nat-hazards-earth-syst-sci-discuss.net/nhess-2019-37/nhess-2019-37-AC1-supplement.pdf

[Figure]

**Fig. 1.** Please see respective rephrased caption of the manuscript Fig. 4 in Author's responses

---

## Author Comment (AC2) · 7 Jun 2019

We would like to thank the Referee#2 for his/her careful review and constructive feedback. We truly believe that the changes suggested by Referee #2 will enhance the quality of the manuscript. Author's responses follow below identified as AR next to the Referee#2 comments.

Referee#2- General comments: The topic of this study is highly relevant, as the Mediterranean area is in need of strategies to cope with weather-related risks, especially drought. The effort to characterize the drought –crop yield losses are therefore valuable. However, in my view, there is a major issue in this manuscript that should be addressed before publication. This is the lack of generalized conclusions. Analyses

[Figure]

are sound but the time and spatial extent and resolution of the data make difficult to extract general conclusions (I agree with Referee 1 on this issue). Even though, the paper would improve a lot if the authors focus on the findings to characterize risk related to drought on these Mediterranean conditions. Findings should be clearly formulated here in a way that the reader can understand what id the added value of the results, beyond the descriptive analysis of the plots at these specific areas.

AR: Thank you for the valuable general comments, in agreement with the two final specific comments. In order to improve the focus in the main findings we propose to follow the suggestion of the Referee #2 of addressing results, discussion and conclusion separately. Moreover, in agreement with Referee #1 and #2 we agree that the spatial and temporal resolutions are not clearly explained in the text and we propose to improve the writing in a revised version.

Referee#2- Specific comments: Abstract: the statement "the estimated conditional probabilities suggest that the likelihood of crop-loss under dry conditions is higher than under non-drought conditions" is pretty obvious for Mediterranean conditions and should not be selected as a main result of the study in the abstract. Please, reformulate to stress the added value of the findings.

AR: We agree to rephrase the main findings of the study focusing on the drought risk levels of wheat and barley. Particularly, the above referred abstract lines 18 and 19 in page 1 may be modified to: "Moreover, the estimated conditional probabilities suggest that the risk of wheat-loss and barley-loss increases 32.53%-32.6% and 31.63%-55.55%, respectively, when drought conditions are below the mild or moderate drought thresholds."

Referee#2: - Pag. 2 line 3 - This is not recent, It has been like this for decades now, as it is main concern in Mediterranean systems

AR: We agree and suggest rephrasing to "From both researcher's and stakeholder's perspective, the management of agricultural drought risk has been a challenging task

for decades, mainly in regions dominated high precipitation variability and recurrent dry and warm episodes, such as the Mediterranean region and in particular the Iberian Peninsula (IP) (Martin-Vide and Lopez-Bustins, 2006; Sousa et al., 2011; Vicente-Serrano et al., 2014)."

Referee#2: - Pag. 3 line 6- Referred to which period?

AR: We believe that the Referee refers to Pag. 2 line 6. The significant negative trends are related to the period 1901–2012 in Páscoa et al. (2017a) and to 1901-2000 in Sousa et al. (2011). In order to be clearer, we propose rephrasing to: "Recent works have found significant negative trends of drought indexes in the IP based on long-term time-series including the entire 20th century, particularly in southern regions (Páscoa et al., 2017a; Sousa et al., 2011), and the expected declining of crop yields due to future warming conditions is being pointed out (Ferrise et al., 2011; Hernández-Barrera and Rodríguez-Puebla, 2017)."

Referee#2: - Pag. 2 lines 14-15 - Given that there are many studies using this methodology at the global scale, I would specify here the issues and flaws, but also the advantages, detected by the authors to enrich the discussion.

AR: Thank you for the kind suggestion. We propose rephrasing to the following in a revised version: "On the other hand, dynamical crop models describing the biological processes are one of the existing tools used to assess crop productivity, e.g. CERES (Crop Environment REsource Synthesis) models (Capa-Morocho et al., 2016; Hlavinka et al., 2010) and AquaCrop (Paredes et al., 2016; Vergni et al., 2015). These crop models are important tools in agrometeorological studies being able to compute irrigation requirements and yield simulations, and have been particularly useful for assessing the impacts of climate change on agricultural productions (Hlavinka et al., 2010). However, such models are limited in their ability to quantify the impact of climate variability on crop yields over larger scales (Estes et al., 2013) and the detailed representation of crop's biophysical interactions requires demanding parameterization settings and input data (Paredes et al. 2014; Giménez et al. 2016; Paredes et al. 2016). Thus, empirical modelling constitutes an alternative to represent the large-scale impacts of drought conditions in the agricultural sector (Vicente-Serrano et al. 2006; Matsumura et al. 2015; Kogan et al. 2015a) requiring lower computation costs than mechanistic modelling (Ferrise et al. 2011; Estes et al. 2013)."

Referee#2: - Pag. 4 line 6 - Somewhere in the paper the low resolution (0.5) of these data and its implications for analysis accuracy should be discussed and handled. Actually, the resolution used is not clear, as in line 19 4 km is mentioned. Please clarify.

AR: The resolution of the gridded datasets of SPEI and remote sensing indices is 0.5° and 4km, respectively. Spatial averages were computed for each provincial cluster, as a result of the exposure analysis performed to ensure that we are addressing the cereals response in the regions dominated by rainfed conditions, which are more vulnerable to droughts (Ribeiro et al., 2018). We agree that the spatial resolution is not clearly explained in the text, as also suggested by the Referee#1. Moreover, Referee#1 also suggested to shorten the data section, hence we repeat below the changes suggested to Referee#1 to rewrite from page 4 line 3 to 27 to the following: "Drought conditions were investigated using two types of indices: the hydro-meteorological drought indicator SPEI and the satellite-based Vegetation Condition Index (VCI) (Kogan 1990), the Temperature Condition Index (TCI) (Kogan 1995) and the Vegetation Health Index (VHI) (Kogan 1995). The monthly drought index SPEI gridded values, with spatial resolution of 0.5°, were computed based on precipitation and temperature values from the Climate Research Unit (CRU TS3.21), using a variety of time scales (1 to 12 months). The weekly global maps of VCI, TCI, and VHI were retrieved at 4km spatial resolution from NOAA's ftp server (ftp://ftp.star.nesdis.noaa.gov/pub/corp/scsb/wguo/data/VHP_4km/geo_TIFF/). While SPEI computation uses climatic water balance anomalies incorporating the role played by the evaporative demand on the occurrence of dry events (Vicente-Serrano et al., 2010), the remote sensing indices characterize the moisture, through the VCI, the temperature induced stress through the TCI and health of vegetation, through the VCI." Page 4 line 15 the sentence "Further spatial averages were computed for each cluster of provinces." and rephrase in Page 4 line 28 "The spatial averages of VCI, TCI, VHI and SPEI were computed for each provincial cluster and used for further modelling of the joint probability between the drought hazard and cereal yield anomalies."

Referee#2: Pag. 4, line 13 – With data from which data source? Please justify with Penman-FAO, in principle more accurate methods, is not used.

AR: The data source used to compute the reference evapotranspiration was obtained from the Climate Research Unit TS3.21 database, which includes monthly values of several climate variables on a global and high-resolution grid (0.5 × 0.5 degrees). Although the Penman-Monteith equation is considered the most robust method for the estimation of the reference evapotranspiration, this method needs a large number of variables, which are not always available. Among the methods that require fewer variables, Beguería et al. (2014) and Vicente-Serrano et al. (2014) recommend the use of the Hargreaves equation in our study area, instead of the Thornthwaite equation. Moreover, this database has been previously used by the authors (Páscoa et al., 2017a) which have also performed a comparison of the reference evapotranspiration using the three above-mentioned methods and the Hargreaves equation have shown a better correlation with the Penman-Monteith than the Thornthwaite method. For these reasons and considering the available data, the Hargreaves method was used in the present work to estimate the reference evapotranspiration.

Referee#2: -Pag. 4 line 29- More details should be provided on these models. This paper should be understandable without reading Ribeiro et al. (2018).

AR: Thank you for your comment, we propose to improve the writing in a revised version to the following: Remove from Page 4 line 15 the sentence "Further spatial averages were computed for each cluster of provinces." Rephrase in Page 4 line 28 -32 and page 5 line 1-2: "Considering the vegetative cycle of wheat and barley, and in accordance

with the results obtained by Ribeiro et al. (2018), the data of VCI, TCI, and VHI used in this work covered the period from week 35 (early September) to week 25 (late June), and data of SPEI covered January to June. The time-scales of SPEI chosen were 1 to 12 months. Spatial averages of all these indicators were computed for each provincial cluster and used for further modelling of the joint probability between the drought hazard and cereal yield anomalies. Stepwise regression models (95% confidence level) were established to select the time scales and months of SPEI, together with the weeks of VCI, TCI, and VHI better related with wheat and barley annual yield (Ribeiro et al. 2018). The selection of the most relevant drought indicator for each cereal and cluster was performed based on the largest absolute value of the standardized regression coefficients, in order to constitute pairs of cereal yield anomalies and drought indicators. Afterwards, for each cereal time series, the joint probability of yield anomalies and the selected drought indicator was estimated."

Referee#2: - Pag. 5 lines 2-6- This information is maybe more appropriate for the introduction. The rest of the section until 2.2 is not clear if it is meant as a summary of the data from Ribeiro et al. (2018) or it is new material with a preliminary analysis id data; if this is the case they should be in the result section.

AR: We agree that this information fits well in the introduction section. The rest of the section is the selection of the most relevant drought indicator for each cereal in each cluster (in order to perform the bivariate models) and as the Referee suggested we agree to move to the results section.

Referee#2: - Section 3.1. More examples for interpreting results in terms of consequences for crop losses-drought weather relationships - Section3.2. Please see my general comments. Maybe separating between results and discussion sections could help. Alternatively, a paragraph of descriptive results should be followed by an interpretation and then extracting generalized statements when possible. Part of this is done in the conclusions section (see below). - Conclusions: In my view, these are not really conclusions but a summary of the study or of the discussion. I would recommend

addressing the discussion separately, and then to reduce and focus the conclusions section.

AR: Thank you for these last 3 suggestions. In a revised version addressing results, discussion and conclusion separately we aim to improve sections 3.1 and 3.2 with more examples of interpreting results in terms of crop losses related to drought conditions and conclusions emphasizing the main findings better pointed out.

References Capa-Morocho, M., Ines, A. V. M., Baethgen, W. E., Rodriguez-Fonseca, B., Han, E. and Ruiz-Ramos, M.: Crop yield outlooks in the Iberian Peninsula: Connecting seasonal climate forecasts with crop simulation models, Agric. Syst., 149, 75–87, doi:10.1016/j.agsy.2016.08.008, 2016. Hlavinka, P., Trnka, M., Eitzinger, J., Smutná, V., Thaler, S., Žalud, Z., Rischbeck, P. and Kr, J.: The performance of CERES-Barley and CERES-Wheat under various soil conditions and tillage practices in Central Europe bei verschiedenen Böden und unterschiedlicher Bodenbearbeitung in Mitteleuropa, , 61(1), 2010. Madadgar, S., AghaKouchak, A., Farahmand, A. and Davis, S. J.: Probabilistic estimates of drought impacts on agricultural pro-duction, Geophys. Res. Lett., 44(15), 7799–7807, doi:10.1002/2017GL073606, 2017. Paredes, P., Rodrigues, G. C., Cameira, M. do R., Torres, M. O. and Pereira, L. S.: Assessing yield, water productivity and farm economic returns of malt bar-ley as influenced by the sowing dates and supplemental irrigation, Agric. Water Manag., doi:10.1016/j.agwat.2016.05.033, 2016. Paredes P, Rodrigues GC, Alves I, Pereira LS (2014) Partitioning evapotranspiration, yield prediction and economic returns of maize under various irrigation management strategies. Agric Water Manag 135:27–39. https://doi.org/10.1016/j.agwat.2013.12.010 Páscoa, P., Gou-veia, C. M., Russo, A. and Trigo, R. M.: Drought trends in the Iberian Peninsula over the last 112 years, Adv. Meteorol., 2017, doi:10.1155/2017/4653126, 2017a. Páscoa, P., Gouveia, C. M., Russo, A. and Trigo, R. M.: The role of drought on wheat yield interannual variability in the Iberian Peninsula from 1929 to 2012, Int. J. Biometeorol., 61(3), 439–451, doi:10.1007/s00484-016-1224-x, 2017b. Vergni,

L., Todisco, F. and Mannocchi, F.: Analysis of agricultural drought characteristics through a two-dimensional copula, Water Resour. Manag., 29(8), 2819–2835, doi:10.1007/s11269-015-0972-4, 2015. Vicente-Serrano et al. Reference evapotranspiration variability and trends in Spain, 1961-2011. Global and Planetary Change, 121, 26-40, https://doi.org/10.1016/j.gloplacha.2014.06.005, 2015. Giménez L, Petillo MG, Paredes P, Pereira LS (2016) Predicting maize transpiration, water use and productivity for developing improved supplemental irrigation schedules in western Uruguay to cope with climate variability. Water 8:1–22. https://doi.org/10.3390/w8070309

Please also note the supplement to this comment:
https://www.nat-hazards-earth-syst-sci-discuss.net/nhess-2019-37/nhess-2019-37-AC2-supplement.pdf

---

## Author Response (AR2)

**Point-by-point reply to the comments**

We would like to thank the Editor and the Reviewers for the valuable remarks and their time. Besides the adjustments requested we have also uniformed the in-text citations with chronological listing. Author's responses follow below identified as AR after the comments from the Editor and Reviewers in bold.

*The manuscript has improved a lot from its previous version. Many thanks for your effort. Anyway, the reviewers have few minor suggestions to improve the manuscript:*

*Page 4, line 7 change CLC by CORINE Land Cover*

AR: Thank you for the recommendation. As suggested CLC was replaced by CORINE Land Cover (2012).

*Page 4, line 12 INE is not necessary you can delete.*

AR: Thank you, as suggested we have deleted "(INE)".

*Page 4, line 14 you can delete Vegetation Condition Index, you have introduced the acronym in page 3 line 26. The same for the Temperature Condition Index*

AR: Thank you for noticing, the suggested change was performed.

*The recent paper of Leng and Hall 2019 could be useful in the introduction and/or discussion section, to provide a world context of the use of copulas in the estimation of yield loss risk under droughts.*

*Leng, G., & Hall, J. (2019). Crop yield sensitivity of global major agricultural countries to droughts and the projected changes in the future. Science of the Total Environment, 654, 811-82*

AR: Thank you for the suggestion. In Introduction section we have added the above reference in page 2 line 18 and introduced the following text in page 3 line 27:

*At the global scale, Leng & Hall (2019) have also used copulas to assess the likelihood of yield loss in response to droughts based on SPI for the a historical period (1961–2016) and future period (2071–2100) under the RCP8.5 emission scenario to investigate future changes in yield loss risk. The authors found that global wheat is more vulnerable to droughts than maize, rice and soybeans, and that global warming is expected to amplify drought-driven yield loss risk.*

Moreover, in Discussion section we have also added the reference in page 11 line 20 and introduced the following text in page 12 line 26:

*Similarly, Leng and Hall (2019) have also used the same copula families and found that from 10 countries 5 of them featured Clayton copulas to fit the joint distribution between wheat production and SPI.*

*Page 11, line 7: please use "hazardous events" instead "hazard events"*

AR: Thank you for the recommendation, the suggested change was performed.

*The final paragraph of page 11, first paragraphs on page 12: Usually discussion does not include calls to Figures or specific data from figures. Instead it should be formulated as general statements, as for instance:*

*Instead*

*"While in Fig. 5 the values of PNE the crop-loss threshold range between 22% (wheat in cluster 1) and 19.2% (barley in cluster 2)…"*

*You could say*

*"While values of PNE the crop-loss threshold were low and similar for wheat in cluster 1 and barley in cluster 2…."*

*etc., (remove also Figure 5 from page 12, line 11)*

AR: Thank you for the valuable suggestion, we have removed the references to Figures in Discussion section and performed the following changes in page 12 line 7:

*While values of PNE the crop-loss threshold were low and similar for wheat in cluster 1 and barley in cluster 2, the values of CPNE the crop-loss threshold during drought years are considerably larger.*

***Conclusions: I appreciate the synthesis effort, but some improvements are still possible. Please remove calls to figures. Do not use the technical names of variables but the represented concepts instead (probability of xxx, standard deviation, variability, etc.). Numerical values (unless really outstanding) should be avoided, use general description (low, moderate, high risk, etc).***

AR: Thank you for the valuable suggestions. We have removed the calls to Figures, substituted PNE and CPNE by unconditional and conditional probabilities of non-exceedance and removed the references to numerical values.

[revised manuscript text omitted]